Reduced gray matter volume in male adolescent violent offenders

Zhang Ying-Dong 1 2
Zhou Jian-Song 1 2
Lu Feng-Mei 3 4
Wang Xiao-Ping xiaop6@csu.edu.cn 1 2
1 Department of Psychiatry & Mental Health Institute of the Second Xiangya Hospital, Central South University , Changsha , Hunan , China
2 National Clinical Research Center on Mental Disorders & National Technology Institute on Mental Disorders, Hunan Key Laboratory of Psychiatry and Mental Health , Changsha , Hunan , China
3 The Clinical Hospital of Chengdu Brain Science Institute, MOE Key Lab for Neuroinformation, University of Electronic Science and Technology of China , Chengdu , Sichuan , China
4 School of Life Science and Technology, Center for Information in Medicine, University of Electronic Science and Technology of China , Chengdu , Cichuan , China
Yuan Tifei
Electronic publication date: 2019 Sep 3
Publication date: 2019
Volume: 7
Electronic Location ID: e7349
Received 2019 May 7; Accepted 2019 Jun 25
Copyright: ©2019 Zhang et al.
Copyright year: 2019
Copyright holder: Zhang et al.
License: This is an open access article distributed under the terms of the Creative Commons Attribution License, which permits unrestricted use, distribution, reproduction and adaptation in any medium and for any purpose provided that it is properly attributed. For attribution, the original author(s), title, publication source (PeerJ) and either DOI or URL of the article must be cited.
License URL: https://creativecommons.org/licenses/by/4.0/

Keywords: Brain structural abnormalities, Discriminate analyses, Violence, ROC, VBM

Funding: National Natural Science Foundation of China 30800368 81371500 81571341 81501637 The MOE (Ministry of Education in China) Project of Humanities and Social Sciences 13YJC190033 This study was supported by National Natural Science Foundation of China (NO. 30800368, 81371500, 81571341 and 81501637), and the MOE (Ministry of Education in China) Project of Humanities and Social Sciences (Project No. 13YJC190033). The funders had no role in study design, data collection and analysis, decision to publish, or preparation of the manuscript.

==============================
Background

Previous studies reported that reduced gray matter volume (GMV) was associated with violent-related behaviors. However, the previous studies were conducted on adults and no study has studied the association between GMV and violent behaviors on adolescents. The purpose of the study was to investigate GMV’s effects in adolescent violent offenders based on a Chinese Han population, which can address the problem of possible confounding factors in adult studies.

Methods

We recruited 30 male adolescent violent offenders and 29 age- and sex-matched healthy controls (HCs). Differences in both whole-brain and GMV were evaluated using voxel-based morphometry (VBM). We assessed the accuracy of VBM using the receiver operating characteristic curve (ROC) and discriminant analysis.

Results

Compared with HCs, the male adolescent offenders showed significantly reduced GMV in five cortical and subcortical brain regions, including the olfactory cortex, amygdala, middle temporal gyrus and inferior parietal lobe in the left hemisphere, as well as the right superior temporal gyrus. Both ROC curve and discriminate analyses showed that these regions had relatively high sensitivities (58.6%–89.7%) and specificities (58.1%–74.2%) with 76.7% classification accuracy.

Conclusions

Our results indicated that reduced volume in the frontal-temporal-parietal-subcortical circuit may be closely related to violent behaviors in male adolescents, which might be an important biomarker for detecting violent behaviors in male adolescents.

Introduction

Violence is a global public health problem that causes great personal sufferings as well as social problems (Krug et al., 2002). The World Health Organization report shows that the global homicide rate in males is significantly higher than in females (13.6 versus 4.0 per 100,000 population) in all age groups (Krug et al., 2002). Among different age groups, adolescents and young adults (15–29 years old) have the highest homicide rates: 19.4 for males and 4.4 for female per 100,000 population (Krug et al., 2002). These reports suggest that the violence problem is more severe in adolescents and young adults, especially males. Moreover, the youth violence has an extremely negative impact on communities and social stability (Greenwood, 2008). Therefore, early identification and management of violent behavior among adolescents not only can save their lives, but also can prevent the development of criminal behaviors in adulthood, thus reducing the burden of crime on society (Greenwood, 2008).

The relationship between abnormal brain structures and violent behaviors has long been noted. It was reported that damage to certain brain areas was related to aggression and violence (Grafman et al., 1996). A longitudinal brain study also revealed that abnormal brain structure was associated with an increased amount of criminal behaviors (McKinlay et al., 2014). In addition, it was reported that the abnormal brain structure and function were key biological risk factors for violence (Rosell & Siever, 2015). Previous identification of persons who were prone to violent behavior was primarily based on personality assessments and electroencephalography. In recent years, the development of noninvasive imaging tools, such as magnetic resonance imaging (MRI), single photon emission computed tomography, and positron emission tomography, has offered new ways for identifying individuals who are at risk of engaging in violent behavior (Leutgeb et al., 2016; Puri et al., 2008).

Evidence from limited structural neuroimaging studies has indicated that reduction of the gray matter volume (GMV) in several brain regions that subserve both emotion processing and behavior regulation (Shi et al., 2016), such as the orbitofrontal cortex (OFC), prefrontal cortex, medial temporal cortex, amygdala, basal ganglia, anterior cingulate cortex, was closely related to violent-related behaviors (Leutgeb et al., 2016; Rosell & Siever, 2015). The brain develops steadily in terms of both structure and function throughout childhood and early adulthood. The developmental trajectory of GMV is complicated and heterogeneous between different brain regions. For example, the GMV in cortex and some subcortical structures (such as the caudate) follows an inverted U-shape developmental course (Giedd & Denker, 2015). In comparison, this developmental trajectory has not been found in other subcortical structures (such as amygdala and hippocampus) (Giedd & Denker, 2015). Moreover, most of the previous studies regarding violence-related structural abnormality were based on adults (Leutgeb et al., 2016; Rosell & Siever, 2015). Only a few studies in children and adolescents have indicated that decreased GMV of several brain regions may be a constant marker for violence-related behaviors (Sterzer & Stadler, 2009; Vloet et al., 2008). For example, lower amygdala volume was reported to be associated with increased violent behaviors from childhood to adulthood and even predicted future violence (Pardini et al., 2014). More importantly, an adolescents based study can avoid possible confounding factors that are commonly seen in adult based studies, such as the shorter duration of illness, medication naïve and lower rates of comorbidities. Thus, studying structural neuroimaging in adolescents has advantages over that in adults.

Moreover, most previous studies on violent behaviors usually were conducted in patients with psychiatric disorders such as conduct disorder (CD), antisocial personality disorder (APD), borderline personality disorder (BPD), attention-deficit hyperactivity disorder (ADHD) or schizophrenia, with aggression, impulsivity, psychopathy or callous-unemotional (CU) traits (Laakso et al., 2002; Schiffer et al., 2011). Only a few previous studies investigated the neuroimaging characteristics of impulsive or aggressive behaviors in healthy subjects (Boes et al., 2009; Boes et al., 2008; Ducharme et al., 2011; Matsuo et al., 2009a; Matthies et al., 2012; Pardini et al., 2014). Furthermore, most previous violence-related brain studies were based on males (Boes et al., 2009; Boes et al., 2008; De Brito et al., 2009; Ducharme et al., 2011; Ermer et al., 2013; Fairchild et al., 2011; Huebner et al., 2008; Sterzer et al., 2007).

The aforementioned violence-brain studies are limited either by using adults or by lacking emphasis on healthy subjects. To fill the gap of the aforementioned GMV-violence studies, we investigated the effects of abnormal brain cerebral structures on violent behaviors and focused the study on healthy male adolescents. More specifically, using voxel-based morphometry (VBM) (Xue, 2016) and region of interest (ROI)-based methods, we analyzed the differences in GMV and white matter volume (WMV) between violent offenders and controls.

Materials & Methods

Participants

A total of 30 male adolescent offenders (aged from 15 to 18 years), who had been convicted of aggressive behavior in court (four murder or manslaughter, 15 intentional injury, and 11 robbery with injury), and 29 age- and gender-matched healthy controls (HCs) were recruited. Male adolescent offenders were recruited from the Youth Detention Centre (YDC), Hunan province, China. All of them had been convicted of either homicide or assault. Male HCs were students, recruited from a middle school in Changsha and further screened by research psychiatrists to exclude subjects with past mental disorders. All participants were right-handed, and had no history of neurological impairments. The substance abusers were excluded based on their urine analysis, self-report and family informant report within the last 3 months.

All interviews were conducted by research psychiatrists. The Chinese version of the Schedule for Affective Disorders and Schizophrenia for School-Age Children Present and Lifetime version (K-SADS-PL) (Shanee, Apter & Weizman, 1997) was used for detecting current and past psychiatric problems according to Diagnostic and Statistical Manual of Mental Disorders (DSM)-IV criteria. We excluded 2 adolescent offenders who were diagnosed with either current or past psychiatric disorders. In addition, we also collected the information of criminal history, psychosocial history, alcohol or other drug use, family history, history of psychiatric, and other medical treatments for each participant.

Signed written informed consent was obtained from all participants and all study procedures were approved by Institutional Review Board of the second Xiangya Hospital, Central South University. 

MRI data acquisition

The structural three-dimensional (3D) T1-weighted MRI images were acquired using a 3.0 T Siemens Vision scanner at the Magnetic Resonance Center of the Hunan Provincial People’s Hospital. A 3D magnetization-prepared rapid-acquisition gradient echo (3D MPRAGE) sequence was used with the following parameter settings: Repetition Time (TR) = 2,000 ms; Echo Time (TE) = 3.36 ms; flip angle = 9°; pixel matrix = 256 × 256, Field of view (FOV) = 256 × 256 mm2, voxel size = 1 × 1 × 1 mm3, and number of slices = 144.

Data processing and statistical analysis

Demographic and clinical characteristics analysis: Two-sample t-tests, chi-square test, and the Fisher exact test were used to analyze the sociodemographic and clinical characteristics of the participants. A p-value less than 0.05 was considered as statistically significant. All analyses were performed using the Statistical Package for Social Sciences (SPSS) version 16.0 (SPSS, Chicago, Ill., USA).

VBM: For the morphometric analysis, the VBM8 (http://dbm.neuro.uni-jena.de/vbm8/), a toolbox of SPM8 (https://www.fil.ion.ucl.ac.uk/spm/software/spm8/) software, was applied with the DARTEL algorithm. For each participant, the origin of the images was first manually aligned to the anterior commissure for a better registration. The structural images were then segmented into gray matter (WM), white matter (WM) and cerebrospinal fluid (CSF) using a standard unified segmentation model in SPM8. Figure 1 shows the segmentation results of one subject. After that, the DARTEL approach was performed for registration, normalization and modulation in the DARTEL space. In details, a new DARTEL template was constructed based on the deformation fields from the segmentation procedure, and all the individual deformation filed map was registered to the new template. The GM images were then normalized to a study-specific template in Montreal Neurological Institute (MNI) space. To preserve the volume of GM, Jacobian determinants are used to modulate the voxel values of GM. After that, the images were smoothed with the Gaussian kernel of 8*8 mm. After spatial preprocessing, the normalized, modulated and smoothed GM and WM images were used for further statistical analyses.

Figure 1 Segmentation results of one typical subject.

These results are obtained in the original space of the images. (A) segmented grey matter (c1X.img); (B) segmented white matter (c2X.img); (C) segmented CSF (c3X.img); (D) segmented skull (c4X.img). CSF, cerebrospinal fluid.

In our study, both whole-brain level and region-of-interest (ROI) level VBM analyses were performed. The whole-brain level analysis is automated and unbiased, making no assumptions of about any regions of particular interest. However, this technique requires a great number of subjects to achieve a statistical significance and therefore changes in smaller structures may be difficult to identify due to the small sample size. To address this problem, a secondary ROI level analysis is often used to test the differences only in the voxels that are deemed of interest by an a priori hypothesis. The ROI level analysis can be used to corroborate the findings of previous whole-brain level studies, or those obtained during the whole-brain analysis. This is of special importance in studies with a small sample size (Husain et al., 2011).

Whole brain analysis: Two sample T-tests were performed to evaluate the significance of the data using the initial threshold of P = 0.001. The p-values were then adjusted using false discovery rate (FDR) correction (with cluster size >100) to correct multiple testing. A FDR-corrected p-value < 0.05 was considered as statistically significant.

ROI analysis: Brain regions that showed significant differences in GMV between two groups in the whole brain analysis were identified as significant ROIs. The main advantage of using a priori ROI test over a whole brain analysis is that it reduces type I errors by narrowing down the statistical tests to only a few ROIs (Husain et al., 2011). For each participant, the mean GMV of each ROI was then extracted using an approach described previously (Santillo et al., 2013). We then used the receiver operating characteristic curve (ROC) analyses to investigate the sensitivity and specificity of the detected abnormalities in GMV, which enabled us to evaluate the discriminative power of these abnormalities in identifying the violent adolescent offenders. In addition, as suggested in one previous study (Jednorog et al., 2014), a further Fisher discriminant analysis was performed to predict whether a participant was a violent offender according to the individual’s GMV for those ROIs which exhibited significant differences (p < 0.001) in GMV between the two groups.

Results

Demographic and clinical characteristics

The results of two-sample T-tests were presented in Table 1. As shown in Table 1, there is no significant difference between the adolescent offenders and HCs in terms of age, total intracranial volume, GMV, and WMV (all p-values >0.05). In comparison, the educational level of the adolescent offenders was significantly lower than that of the HCs (7.5 ± 2.4 vs 10.0 ± 0.0 years; t =  − 5.10; p-value < 0.01), which was adjusted in the following analyses. The educational difference is due to that most of the adolescent offenders had dropped out from the junior high school, while all HCs were students from senior schools.

Table 1 Sociodemographic and clinical characteristics (mean ± S.D.) in adolescent offenders and healthy controls (two-sample t-tests).

	HC(N = 29)	AO(N = 30)	Statistic	P-value	
Age, years	17.6 ± 0.5	17.7 ± 0.8	t = 0.45	0.66	
Literacy, years	10.0 ± 0.0	7.5 ± 2.4	t =  − 5.10	0.00	
TIV, mm3	1335 ± 86	1348 ± 107	t = 0.45	0.65	
GMV, mm3	633 ± 39	629 ± 46	t =  − 0.36	0.72	
WMV, mm3	462 ± 38	473 ± 50	t = 0.84	0.40	
Notes.

Abbreviations S.D. standard deviation

HC healthy controls

AO adolescent offenders

TIV total intracranial volume

GMV gray matter volume

WMV white matter volume

Whole brain analysis

The result of whole-brain analysis was presented in Table 2. Compared to the HCs, the adolescent offenders showed significantly lower GMV (p < 0.05, FDR corrected) in the olfactory cortex, amygdala, middle temporal gyrus and inferior parietal lobe in the left hemisphere, as well as the right superior temporal gyrus (Fig. 2). In comparison, there was no significant difference in GMV of the whole brain between the offenders and HCs. Similarly, no significant difference in WMV was observed between the two groups.

Table 2 Regions with smaller gray matter volume in adolescent offenders (N = 31) compared with healthy controls (N = 29), measured using voxel-based morphometry (corrected false discovery rate at P < 0.05).*

Regions	BA	Stereotactic coordinates (mm)	Cluster size(mm3)	T	P-value	
		x	y	z				
L OC	11	−17	12	−14	232	−4.68	0.000**	
L AMG	34	−26	3	−15	191	−6.50	0.014	
L MTG	21	−65	−1	−11	103	−5.12	0.047	
L IPL	39	−39	−52	54	130	−4.43	0.000**	
R STG	21	40	0	−15	104	−4.72	0.000**	
Notes.

Abbreviations BA Brodmann’s area

T intensity

L left

R right

OC olfactory cortex

AMG amygdala

MTG middle temporal gyrus

IPL inferior parietal lobe

STG superior temporal gyrus

* The duration of education was select as the covariant.

** Uncorrected false discovery rate.

Figure 2 Brain voxel-based morphometry showing lower gray matter volume in male violent adolescent offenders superimposed on a T1-weightedtemplate (violent adolescent offenders vs. controls; two-sample t-tests).

These brain areas include the left olfactory cortex (A, B), the left amygdala (C, D), the left middle temporal gyrus (C, D), the left inferior parietal lobe (E), and the right superior temporal gyrus (F). The color bar indicates the t value of the between-group analysis.

ROI Analysis

Five ROIs were identified as significant based on the whole-brain analysis, including the left olfactory cortex, left amygdala, left middle temporal gyrus, left inferior parietal lobe, and the right superior temporal gyrus. A further ROC analysis was conducted on these ROIs and the results were presented in Table 3. The sensitivity, specificity and area under the curve (AUC) values were 58.6%, 67.7% and 0.642 for the left olfactory cortex; 86.2%, 74.2% and 0.834 for the left amygdala; 79.3%, 58.1% and 0.721 for the left middle temporal gyrus; 86.2%, 74.2% and 0.834 for the left inferior parietal lobe; and 89.7%, 64.5% and 0.790 for the right superior temporal gyrus. It was suggested that the AUC values can be interpreted as excellent if AUC ≥ 0.90; good if 0.90 > AUC ≥ 0.80; fair if 0.80 > AUC ≥ 0.70; poor if 0.70 > AUC ≥ 0.60; and no effect if AUC < 0.60 (Linden, 2006). Hence, our results suggested that except for the left olfactory cortex, GMV reduction of each of the other four ROIs represented a fair or good biomarker for violence in juveniles.

Table 3 Area under the curve (AUC) details for the five brain regions of interest showing significant differences in gray matter volume between male adolescent offenders and male age-matched controls.

	L OC	L AMG	L MTG	L IPL	R STG	
Sensitivity%	58.6	86.2	79.3	86.2	89.7	
Specificity%	67.7	74.2	58.1	74.2	64.5	
AUC	0.642	0.834	0.721	0.834	0.790	
Notes.

Abbreviations L left

R right

OC olfactory cortex

AMG amygdala

MTG middle temporal gyrus

IPL inferior parietal lobe

STG superior temporal gyrus

AUC area under the curve

These five ROIs were then used for a discriminant analysis. The overall classification accuracy was 76.7%: 75.9% accuracy for the HC group versus 77.4% accuracy for violent adolescent offenders. The validity of using GMV to distinguish two groups was supported by Wilk’s lambda = 0.57 (df = 4, p < 0.001).

Discussion

To our best knowledge, this study represented the first attempt to use the VBM method for identifying brain structural abnormalities in a cohort of male violent adolescents with no history of mental disorders. We found that male adolescent offenders had a reduced GMV in the olfactory cortex, amygdala, middle temporal gyrus and inferior parietal lobe in the left hemisphere, and in the right superior temporal gyrus than male HCS. These findings provide evidence that male violent adolescent offenders exhibit reductions of GMV in several brain regions.

The pathophysiology of violence-related behaviors has not been understood clearly. Several lines of studies showed that violent behaviors were related with structural or functional abnormalities in several brain regions that subserve both emotion processing and behavior regulation (Leutgeb et al., 2016; Rosell & Siever, 2015). The amygdala is a subcortical structure of limbic system that plays an essential role in the integration of a wide range of sensory and motivationally salient stimuli, as well as in transmission of this information to various cortical and subcortical regions. These processes underlie the amygdala’s essential role in fear mediating, defensive reactions, emotional learning, and motivation. The amygdala has long been considered as the most important neural region for violence-related behaviors (Lane, Kjome & Moeller, 2011; Perathoner, Cordero-Maldonado & Crawford, 2016). Similar to the finding in our current study, several previous studies found a decreased amygdala volume in juvenile subjects prone to violence than their healthy peers (Fairchild et al., 2011; Huebner et al., 2008; Sterzer et al., 2007; Stevens & Haney-Caron, 2012). One longitudinal study reported that a lower amygdala volume was a significant factor for detecting increased violent behaviors from childhood to adulthood (Pardini et al., 2014). Further, the frontal cortex, especially the OFC, exhibits the most robust reciprocal anatomical connections with the amygdala, which receives limbic inputs from amygdala and other medial temporal areas as well as sensory inputs (Leutgeb et al., 2016). Thus, frontal cortex may integrate sensory information with affective signals, inhibitory control signals from other areas and is considered as the trigger point of violence-related behaviors (Lane, Kjome & Moeller, 2011; Rosell & Siever, 2015). Several previous studies reported decreased GMV of OFC in subjects prone to violence (Bertsch et al., 2013; Boes et al., 2009; De Oliveira-Souza et al., 2008; Huebner et al., 2008; Laakso et al., 2002; Tiihonen et al., 2008), negative correlation between GMV of OFC and impulsive (Kumari et al., 2009; Matsuo et al., 2009a), aggressive (Gansler et al., 2009) or psychopathic (Ermer et al., 2012; Ermer et al., 2013) level. These results support our finding of decreased GMV of olfactory cortex, as part of the OFC, in violent adolescents. In addition, the temporal cortex and the parietal cortex, both of which are closely connected to the limbic system, are important for regulating the affective nature of interpersonal experiences and play a pivotal role in the development of emotional behavior (Soderstrom et al., 2002). Previous studies showed decreased GMV of several regions of the temporal or the parietal lobe, such as superior temporal cortex (Bertsch et al., 2013; De Oliveira-Souza et al., 2008; Muller et al., 2008; Stevens & Haney-Caron, 2012), middle temporal cortex (Bertsch et al., 2013), inferior temporal cortex (Bertsch et al., 2013; Gregory et al., 2012; Stevens & Haney-Caron, 2012), hippocampus (Barkataki et al., 2006; Huebner et al., 2008; Stevens & Haney-Caron, 2012; Yang et al., 2010), parahippocampal gyrus (Bertsch et al., 2013; Stevens & Haney-Caron, 2012; Yang et al., 2010), temporal pole (Bertsch et al., 2013), inferior parietal cortex (Tiihonen et al., 2008), postcentral cortex (Bertsch et al., 2013; Stevens & Haney-Caron, 2012; Tiihonen et al., 2008), angular gyrus (Puri et al., 2008) and supramarginal gyrus (Puri et al., 2008; Stevens & Haney-Caron, 2012) in subjects with a tendency toward violence. Also, we found low GMV of three regions in the temporal or parietal lobe, the middle temporal gyrus, the superior temporal gyrus and the inferior parietal lobe in adolescent violent offenders, which were consistent with these previous studies. Our results and previous literatures indicate that reduced GMV in the frontal-temporal-parietal-subcortical circuit may lead to difficulties in suppressing expressions of emotion, which may further lead to inappropriate, or even violent behaviors.

Low GMV of several areas in the frontal-temporal-parietal-subcortical circuit was found to be a marker for violence-related behaviors both in juveniles (Boes et al., 2009; Boes et al., 2008; Bussing et al., 2002; Ducharme et al., 2011; Ermer et al., 2013; Fairchild et al., 2011; Huebner et al., 2008; Kruesi et al., 2004; Sterzer et al., 2007) and in adults (Barkataki et al., 2006; Bertsch et al., 2013; De Oliveira-Souza et al., 2008; Ermer et al., 2012; Gansler et al., 2009; Gregory et al., 2012; Kumari et al., 2009; Laakso et al., 2002; Matsuo et al., 2009a; Matsuo et al., 2009b; Matthies et al., 2012; Muller et al., 2008; Pardini et al., 2014; Puri et al., 2008; Raine et al., 2000; Tiihonen et al., 2008; Yang et al., 2010; Yang et al., 2005; Zhang et al., 2013).

We evaluated the discriminative power of both GMV and the identified brain regions in classifying violent offenders and controls.The ROC analysis with VBM on each region revealed that reduced GMV in four regions (left amygdala, left middle temporal lobe, left inferior parietal lobe, and right superior temporal gyrus) represented either a fair or a good biomarker of violence in juveniles. Further discriminant analysis indicated that reduced GMVs of the five regions could predict whether a subject was a violent offender with 76.7% accuracy, suggesting that reduced GMV of the five regions could be regarded as a biomarker of violent behavior in male adolescents. The high classifyication accuracy demonstrates that abnormal brain structure may be a partial cause of violent behaviors.

Limitations

There are several limitations in this study. (1) This study was conducted solely in male adolescents. Thus, our findings cannot be generalized to females, adults, and children younger than 14 years old. (2) In our study setting, we only have two classes: adolescents with violent behaviors versus adolescents without violent behaviors. That is to say, different types of violent behaviors are treated as the same class, i.e., violent behavior. As a result, only classification experiment analysis was conducted. Nevertheless, it is feasible to quantify different types of violent behaviors by assigning different scores according to the severity. Further correlation analysis based on the scores may help uncover more findings. (3) The study is based on a small cohort of 59 participants. Additional experiments on other independent samples/cohort can help further validate our findings. However, as mentioned in our introduction, little work has been done to investigate the association of abnormal brain with violence tendency in healthy male adolescents. Therefore, at the moment we are not able to find an independent cohort to validate the findings. (4) Several other factors, e.g., stress and personality traits, are also related to violence tendency. However, the information of these personality traits were not available in the current cohort, which limited the conclusion of our work.

Conclusions

We found a reduced GMV in five different brain regions in male adolescent offenders compared to that in male adolescent HCS. Moreover, our analyses verified the validity and practicality of using structural neuroimaging analyses to distinguish violent adolescent offenders and non-violent adolescents in males. Specifically, VBM technique is helpful for characterizing violent male adolescents. In addition, the findings in this study suggest that reduced volume in the frontal-temporal-parietal-subcortical circuit may be closely associated with violent behaviors in male adolescents, and thus could represent an important potential biomarker for detecting violent behavioral tendencies in male adolescents.

Supplemental Information

Data S1 Raw Data: Magnetic resonance imaging (MRI) data

This can be read and processed by using the tool SPM (version: SPM12; available at: https://www.fil.ion.ucl.ac.uk/spm/software/download).

Click here for additional data file.

Additional Information and Declarations

Competing Interests

Author Contributions

Human Ethics

Data Availability

The authors declare there are no competing interests.

Ying-Dong Zhang performed the experiments, analyzed the data, contributed reagents/materials/analysis tools, prepared figures and/or tables, authored or reviewed drafts of the paper, approved the final draft.

Jian-Song Zhou and Feng-Mei Lu authored or reviewed drafts of the paper, approved the final draft.

Xiao-Ping Wang conceived and designed the experiments, contributed reagents/materials/analysis tools, authored or reviewed drafts of the paper, approved the final draft.

The following information was supplied relating to ethical approvals (i.e., approving body and any reference numbers):

All study procedures were approved by Institutional Review Board of the second Xiangya Hospital, Central South University.

The following information was supplied regarding data availability:

The raw data is available as a Supplemental File.

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
