# Peer review of "Reduced gray matter volume in male adolescent violent offenders"

_PeerJ, doi:10.7717/peerj.7349_

## Round 0.1 · original submission · Major Revisions

Please revise the paper according to the comments.

Reviewer 1 ·

Basic reporting

'no comment'

Experimental design

'no comment'

Validity of the findings

'no comment'

Additional comments

1. The author mentioned that most of the previous studies regarding violence-related structural abnormality were carried out in adults (page 7 line 80) . I am wondering is there any difference between adults and adolescents on GMV.

2. The English language should be improved to ensure the audience can clearly understand your text. For example:
Furthermore, the majority of previous violence-related studies were carried out on male(page8 line 99)

3. The author mentioned this study focus on specific regions (page 8 line 103). How to define “specific”? why the author focus on this “specific regions”?

4. Did author consider the comorbidity among adolescent offenders (CD, ADHD,PD, BD) in this study? Since the comorbidity cold be act as confounding factor.

5. The author mentioned“it is reasonable to confer that reduced GMV in the frontal-temporal-parietal-subcoritcal ……….. thus resulting in inappropriate, or even violent behaviors”.
6. This is a cross-section study, so the conclusion should be interpreted with some caution.

7. There are other factors related to violent behaviors, such as stress and personality traits. Author need to address these factors in the limitation.

·

Basic reporting

Professor Wang and his team reported that reduced GMV in the frontal-temporal-parietal- subcortical circuit related to violent behaviors in 30 male adolescents. This study provided important clue for us to investigate the biomarker of violent behaviors, especially in the forensic psychiatry. As a whole, the findings of this study provide useful biomarkers for we investigating the violets behaviors. Although the sample is relatively small, from my opinion, we needed this kind study to explore the biomarker to help the psychiatrist, especially for the psychiatrist who act as a forensic psychiatry expert.

Experimental design

None comments about the design.

Validity of the findings

The findings provided usueful clues for us to investigate the biomarker of violent behaviors.

Additional comments

I provide some comments as bellow for the author to consider,

1. In the section of introduction, provide a hypothesis about you study will enhance the celerity for us understand your study.
2. In the section of Methods , please detailed describe the "study-specific template in Montreal Neurological Institute (MNI) space" , because the samples in your study are adolescents, it need specific template.

3. In the section of results, please add the scatter plot of sensitivity and specificity.

Reviewer 3 ·

Basic reporting

Line 53-54,56-57,102: English language need improve, some typos

In the method section, when the authors did those comparisons, did they control for the potential covariates, such as the total intracranial volume — the total value of brain white matter, gray matter and CSF they listed in table 1.

Results from line 208 to 209 were repeated as they were mentioned in line 205-208.

Figure 2 is not quite clear such as we don’t even know the left or right side of the brain.

More information should be showed in table 1, such as the different kinds of aggressive behavior as the authors mentioned a little bit in line 111 or other demographic information of all participants.

Experimental design

no comment.

Validity of the findings

no comment.

Additional comments

As the authors mentioned, several previous studies reported decreased GMV of OFC in subjects prone to violence (Bertsch et al., 2013), negative correlation between GMV of OFC and impulsive (Kumari et al., 2009), aggressive (Gansler et al., 2009) or psychopathic (Ermer et al., 2012, 2013) level. I was wondering whether they have some scales for evaluating the offending behaviors or symptoms of those adolescents? If so, that would be interesting to see whether those finding of brain regions were correlated with those scores or symptoms. If not, they may add it in their study in the future.

That would be great if the authors are able to have validation of those finding in some other independent sample set.

---

## Round 0.2 · accepted · Accept

Thanks for the efforts in improving the manuscript.

·

Basic reporting

The revision manuscript is better. The answer is better.

Experimental design

Better

Validity of the findings

better

Additional comments

The revision manuscript is better,about the question I commentted, the author had answered , the first MNI, I aim to suggest she/he using three deminsional method, is will acquried a better viusal effect. However, this did not influence the result. About the scatter polt, figure , it will demonstrat the relatinship of brain alterations and syptoms. You reported in your mamuscript. I suggestion is that ,I like Figure to demonstrated your result, this is my style , dont infulecnce your results.

In summary, I agree the answer of my comments, I agree to accept this mamuscript.

Reviewer 3 ·

Basic reporting

no comment

Experimental design

no comment

Validity of the findings

no comment

Additional comments

The revised manuscript is improved.